# EPIC: Inferring relevant cell types for complex traits by integrating genome-wide association studies and single-cell RNA sequencing

**Rujin Wang**[1], **Dan-Yu Lin**[1,2]*, **Yuchao Jiang**[1,2,3]*

**1** Department of Biostatistics, Gillings School of Global Public Health, University of North Carolina, Chapel Hill, North Carolina, United States of America, **2** Lineberger Comprehensive Cancer Center, University of North Carolina, Chapel Hill, North Carolina, United States of America, **3** Department of Genetics, School of Medicine, University of North Carolina, Chapel Hill, North Carolina, United States of America

* lin@bios.unc.edu (D-YL); yuchaoj@email.unc.edu (YJ)

**Data Availability Statement:** GWAS summary statistics are downloaded from public repositories listed in Table A in S1 Text. Genotypes from the 1000 Genomes Project reference panel are

## Abstract

More than a decade of genome-wide association studies (GWASs) have identified genetic risk variants that are significantly associated with complex traits. Emerging evidence suggests that the function of trait-associated variants likely acts in a tissue- or cell-type-specific fashion. Yet, it remains challenging to prioritize trait-relevant tissues or cell types to elucidate disease etiology. Here, we present EPIC (cEll tyPe enrIChment), a statistical framework that relates large-scale GWAS summary statistics to cell-type-specific gene expression measurements from single-cell RNA sequencing (scRNA-seq). We derive powerful gene-level test statistics for common and rare variants, separately and jointly, and adopt generalized least squares to prioritize trait-relevant cell types while accounting for the correlation structures both within and between genes. Using enrichment of loci associated with four lipid traits in the liver and enrichment of loci associated with three neurological disorders in the brain as ground truths, we show that EPIC outperforms existing methods. We apply our framework to multiple scRNA-seq datasets from different platforms and identify cell types underlying type 2 diabetes and schizophrenia. The enrichment is replicated using independent GWAS and scRNA-seq datasets and further validated using PubMed search and existing bulk case-control testing results.

## Author summary

Genome-wide association studies (GWASs) have yielded genetic variants associated with various complex traits. Emerging evidence suggests that the function of trait-associated variants likely acts in a tissue- or cell-type-specific fashion. For many complex traits, however, the specific cell or tissue types leading to risk are unknown. Recent advances of single-cell RNA sequencing (scRNA-seq) provide unprecedented opportunities, alongside challenges, to systematically investigate the cell-type-specific enrichment of GWAS risk variants. We propose EPIC, a statistical framework that relates large-scale GWAS summary statistics to cell-type-specific transcriptomic measurements from scRNA-seq data to

available at https://ctg.cncr.nl/software/magma. Bulk RNA-seq and scRNA-seq data are downloaded from GTEx v8 at http://www.gtexportal.org. ScRNA-seq read counts from two pancreatic islet studies are publicly available with accession GSE84133 (https://www.ncbi.nlm.nih.gov/geo/query/acc.cgi?acc=GSE84133) and E-MTAB-5061 (https://www.ebi.ac.uk/arrayexpress/experiments/E-MTAB-5061). We obtain a list of human housekeeping genes from the Housekeeping and Reference Transcript Atlas at https://housekeeping.unicamp.br. EPIC is compiled as an open-source R package available at https://github.com/rujinwang/EPIC. Scripts used for analyses carried out in this paper are deposited in the GitHub repository.

**Funding:** This work was supported by the National Institutes of Health (NIH) grant P01 CA142538 (to D.L. and Y.J.), R35 GM138342 (to Y.J.), and R01 HG009974 (to D.L.). The funders had no role in study design, data collection and analysis, decision to publish, or preparation of the manuscript.

**Competing interests:** The authors have declared that no competing interests exist.

prioritize trait-relevant cell types. We use known trait-relevant tissues and cell types as ground truths for benchmark, adopt independent GWAS and scRNA-seq datasets for reproducibility, and refer to PubMed keyword search and existing case-control studies for validation. Such an integrative analysis helps elucidate the underlying cell-type-specific disease etiology and prioritize important risk variants.

## Introduction

Many years of genome-wide association studies (GWASs) have yielded genetic risk variants associated with complex traits and human diseases. Emerging evidence suggests that the function of trait-associated variants likely acts in a tissue- or cell-type-specific fashion [1,2]. Recent advances in single-cell RNA sequencing (scRNA-seq) enable characterization of cell-type-specific gene expression and provide an unprecedented opportunity to systematically investigate the cell-type-specific enrichment of GWAS polygenic signals [3–6]. There is a pressing need to develop a statistically rigorous and computationally scalable analytical framework to integrate large-scale genome-wide association studies (e.g., the UK Biobank [7]) and high-dimensional scRNA-seq efforts (e.g., the Human Cell Atlas [8]). Such an integrative analysis helps elucidate the underlying cell-type-specific disease etiology and prioritize important functional variants.

Several methods [9–13] have been developed to integrate scRNA-seq data with GWAS summary statistics to prioritize trait-relevant cell types. One set of methods, including Roly-Poly [9] and LDSC-SEG [11], develops models on the single-nucleotide polymorphism (SNP) level and derives SNP-wise annotations from the transcriptomic data. RolyPoly adopts a polygenic model, and the effect sizes of all SNPs associated with a gene have a covariance that is a linear combination of the gene expressions across all cell types. RolyPoly, therefore, captures the effect of the cell-type-specific gene expression on the covariance of GWAS effect sizes, which can be computationally intensive. LDSC-SEG also constructs SNP annotations from cell-type-specific gene expressions and then carries out a one-sided test using the stratified LD score regression framework [11,14,15]. It tests whether trait heritability is enriched in regions surrounding genes that have the highest cell-type-specific expression.

Another set of methods, such as CoCoNet [12] and MAGMA [3,10,16,17], does not devise the SNP-level framework. These methods first derive gene-level association statistics since this more naturally copes with the gene-level expression measurements; they then prioritize risk genes in a specific cell type. Specifically, CoCoNet models gene-level test statistics as a function of the cell-type-specific adjacency matrices, which are inferred from the gene expression measurements. While CoCoNet is the first method to evaluate the gene co-expression networks, its rank-based method does not allow hypothesis testing due to the strong correlation among gene co-expression patterns constructed from different cell types. Like CoCoNet, MAGMA and MAGMA-based approaches also begin by combining SNP-level GWAS summary statistics into gene-level statistics. This step is followed by a second "gene-property" analysis, where the cell-type-specific gene expressions are regressed against the genes' GWAS test statistics. The various versions of the methods adopt different ways to select genes, transform the outcome and predictor variables, and include different sets of additional covariates [3,10,16,17]. While MAGMA-based methods have been successfully used in several studies [18–20], Yurko et al. [21] examined the statistical foundation of MAGMA, and they identified an issue: type I error rate is inflated because the method incorrectly uses the Brown's approximation when combining the SNP-level $p$-values. MAGMA's implementation replaces the SNP-SNP correlation

coefficient with its square, which serves as a rough correction yet masks the true LD structure. A rigorous testing framework that is powerful and controls for false positives is needed.

When modeling on the gene level, one needs to account for the gene-gene correlations. RolyPoly ignores proximal gene correlations but implements a block bootstrapping procedure as a correction. MAGMA approximates the gene-gene correlations as the correlations between the model sum of squares from the second-step gene-property analysis. However, the gene-gene correlation of the effect sizes should be a function of the LD scores (i.e., the correlations between the SNPs within the genes). CoCoNet does not take account of this either, instead using LD information only to calculate the gene-level effect sizes and assuming that gene-gene covariance is a function solely of gene co-expression. A statistically rigorous and computationally efficient method to derive the gene-gene correlation structure while incorporating the SNP-level LD information is needed.

These existing methods either focus on common variants (e.g., RolyPoly and LDSC-SEG) or do not differentiate between common and rare variants (e.g., MAGMA with only summary statistics) due to the limited statistical power for rare variants. While methods for rare-variant association analysis have been developed (e.g., sequence kernel association test [22] and burden test [23]), to our best knowledge, no methods are currently available to detect cell-type-specific enrichment of GWAS risk loci using summary statistics for both common and rare variants.

Here, we propose EPIC, a statistical framework to identify trait-relevant cell types by integrating GWAS summary statistics and cell-type-specific gene expression profiles from scRNA-seq. We adopt gene-based generalized least squares to identify enrichment of risk loci. For the prioritized cell types, EPIC further carries out a gene-specific influence analysis to identify significant genes. Compared to existing methods, EPIC's main advantages include the following: (i) a statistical framework for association testing based on multivariate GWAS summary statistics that is powerful and controls for type I error; (ii) separate and joint modeling of common and rare variants when integrating GWAS and single-cell sequencing data; (iii) a rigorous and scalable regression framework that accounts for gene-gene correlations; and (iv) a cell-type- and gene-specific influential testing scheme to identify genes and gene sets that are relevant to the significant enrichment. We demonstrate EPIC on multiple tissue-specific bulk RNA-seq and scRNA-seq datasets, along with GWAS summary statistics of four lipid traits, three neuropsychiatric disorders, and type 2 diabetes, and successfully replicate and validate the prioritized tissues and cell types. Together, EPIC's integrative analysis of cell-type-specific expressions and GWAS polygenic signals help to elucidate the underlying cell-type-specific disease etiology and prioritize important functional variants. EPIC is compiled as an open-source R package available at https://github.com/rujinwang/EPIC.

## Results

### Overview of methods

The goal of EPIC is to identify disease- or trait-relevant cell types. An overview of the framework is outlined in Fig 1. EPIC takes as input single-variant summary statistics from GWAS, which is used to aggregate SNP-level associations into genes, and gene expression datasets from scRNA-seq data. An external reference panel is adopted to account for the linkage disequilibrium (LD) between SNPs and genes. We first perform gene-level testing based on GWAS summary statistics from the single-variant analysis. The multivariate statistics for both common and rare variants can be recovered using covariance of the single-variant test statistics, which can be estimated from either the raw genotype matrix or a reference panel (e.g., the 1000 Genomes Project). We then develop a gene-based regression framework that can prioritize trait-relevant cell types from gene-level test statistics and cell-type-specific omics profiles

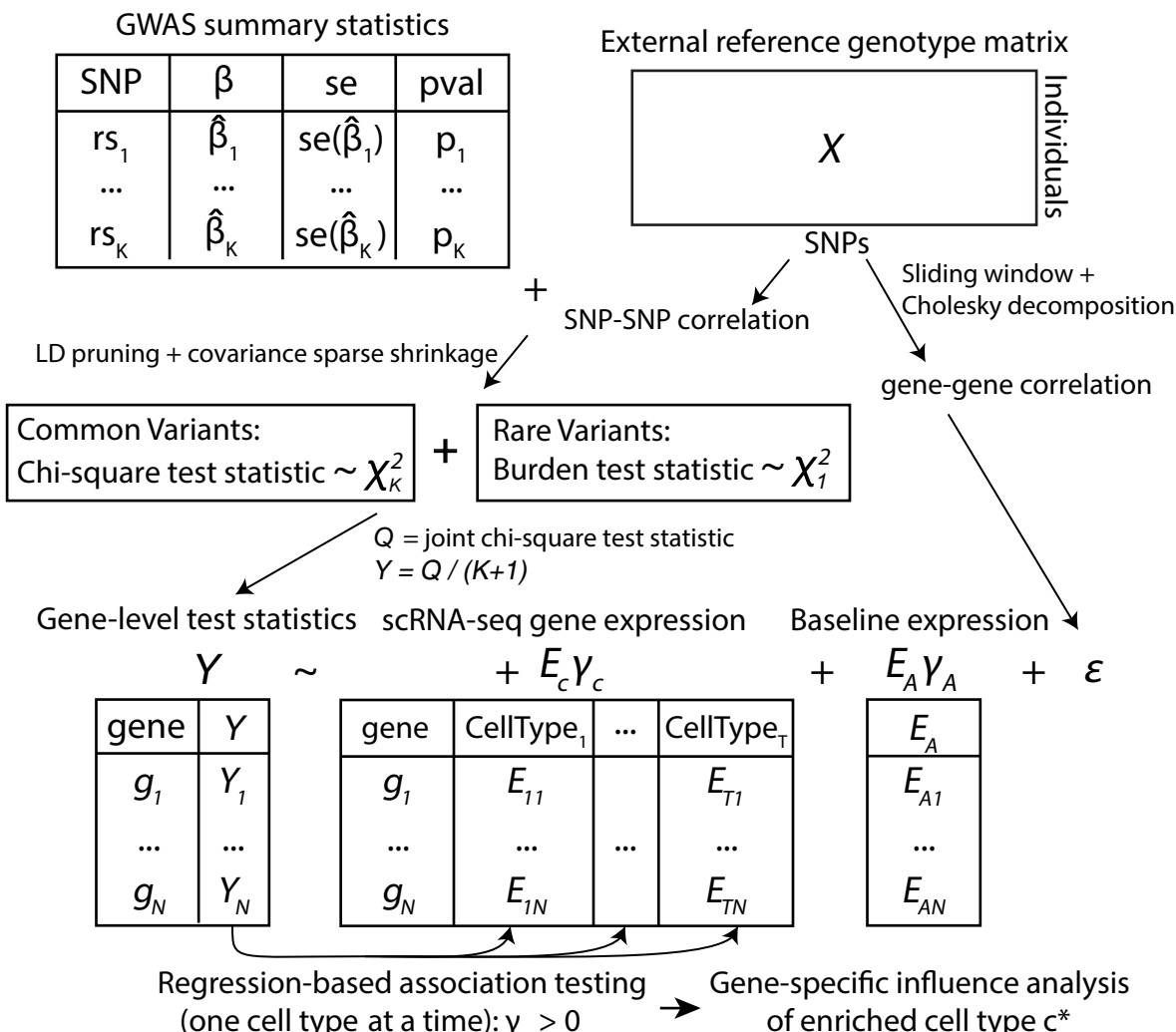

**Fig 1. Overview of EPIC framework.** EPIC starts from GWAS summary statistics and an external reference panel to account for LD structure. To ensure that the correlation matrix is well-conditioned, EPIC adopts the POET estimators to obtain a sparse shrinkage correlation matrix. EPIC performs LD pruning, computes the gene-level chi-square statistics for common variants, and calculates burden test statistics for rare variants. EPIC then integrates gene-level association statistics with transcriptomic profiles and prioritizes trait-relevant cell types using a regression-based framework while accounting for the gene-gene correlation structure.

while accounting for gene-gene correlations due to LD. The underlying hypothesis is that if a particular cell type influences a trait, then more of the GWAS polygenic signals would be concentrated in genes with greater cell-type-specific gene expression. For significantly enriched cell type(s), we further carry out a gene-specific influence analysis to identify genes that are highly influential in leading to the significance of the prioritized cell type. Refer to the Methods section for methodological and algorithmic details.

## EPIC's gene-level association testing

We first performed the gene-level chi-square association test with the shrinkage estimators and sliding-window approach using GWAS summary statistics for eight diseases and traits (four lipid traits [24], three neuropsychiatric disorders [25–27], and type 2 diabetes (T2Db) [28]; Table A in S1 Text). We benchmarked against the aforementioned method, MAGMA, as

well as ACAT [29], a method that combines correlated SNP-level p-values via Cauchy combination test and does not make any assumptions on the direction of the effects. In Table B in S1 Text, we summarized a list of genes that have been previously reported to be associated with traits [24–28]; for these sets of well-characterized trait-associated genes, EPIC returned more significant *p*-values compared to MAGMA and ACAT, especially when the common and rare variants are jointly modeled. On the genome-wide scale, the quantile-quantile (Q-Q) plots of gene-level *p*-values demonstrated EPIC's elevated power (Fig A in S1 Text), and EPIC detected more significant genes than MAGMA after Bonferroni correction (Fig B in S1 Text). For the significantly associated genes detected by EPIC, but not by MAGMA, we performed functional enrichment analysis using DAVID [30] and identified gene ontology (GO) biological processes directly relevant to the traits of interests (Table C in S1 Text).

Importantly, we further reported gene-level association testing results for a set of housekeeping genes [31] and demonstrated that, while powerful, EPIC also controlled for type I error (Fig C in S1 Text). Notably, for psychiatric diseases and type 2 diabetes (T2Db), all three methods that we benchmarked (i.e., EPIC, MAGMA [10,17], and ACAT [29]) seemingly show inflated false positive rates–we focused on the top five significantly associated genes and confirmed that they had been previously reported to be trait relevant (see Fig C in S1 Text for details). This indicates that part of the housekeeping genes, while constitutively expressed to maintain cellular functions, can still be associated with complex traits. Therefore, for all empirical studies, we generated Q-Q plots for genes with *p*-values from 0.05 to 1, excluding potential trait-relevant genes; we showed that across all traits EPIC's *p*-values in this range are uniformly distributed (Fig A in S1 Text).

## Inferring trait-relevant tissues using tissue-specific RNA-seq

As a proof of concept, we applied EPIC to identify trait-relevant tissues by integrating the gene-level testing statistics from the previous section with tissue-specific transcriptomic profiles from the Genotype-Tissue Expression project (GTEx) v8 [32] (Table A in S1 Text). The GTEx consortium consists of bulk-tissue gene expression measurements of 17,382 samples from 54 tissues across 980 postmortem donors; after sample-specific quality controls, we obtained gene expression profiles of 45 tissues, averaged across samples (Table A in S1 Text). For subsequent analyses, we focused on a set of 8,708 genes with tissue specificity scores greater than 5 (Note A in S1 Text).

All four lipid traits are significantly enriched in the liver (Fig 2), which plays a key role in lipid metabolism. The small intestine was marginally enriched for TC–it has been shown that the small intestine plays an important role in cholesterol regulation and metabolism [33]. In addition, the adipose tissues, which have also been shown to regulate lipid metabolism [34], were identified as being significantly enriched by both EPIC and MAGMA. Both LDSC-SEG and RolyPoly suffer from low power, although the liver was one of the top-ranked tissues for the lipid traits. The pancreas and the liver were prioritized as the T2Db-relevant tissues by EPIC, while MAGMA yielded significant results in the pancreas as well as the stomach (Fig 3A). RolyPoly identified the pancreas as the second most relevant tissue; LDSC-SEG reported the liver as the only significantly enriched tissue (Fig 3A). Finally, neuropsychiatric disorders exhibited strong brain-specific enrichments, as expected. The frontal cortex of the brain was detected as being the most strongly enriched for SCZ, BIP, and SCZBIP (Fig 4A). The pituitary also demonstrated strong enrichment signals with SCZ and SCZBIP, while the spinal cord was found to be an irrelevant tissue with these three neuropsychiatric disorders. In comparison, LDSC-SEG identified part of the brain tissues as trait-relevant, while RolyPoly failed to return enrichment in any of the brain tissues (Fig 4A).

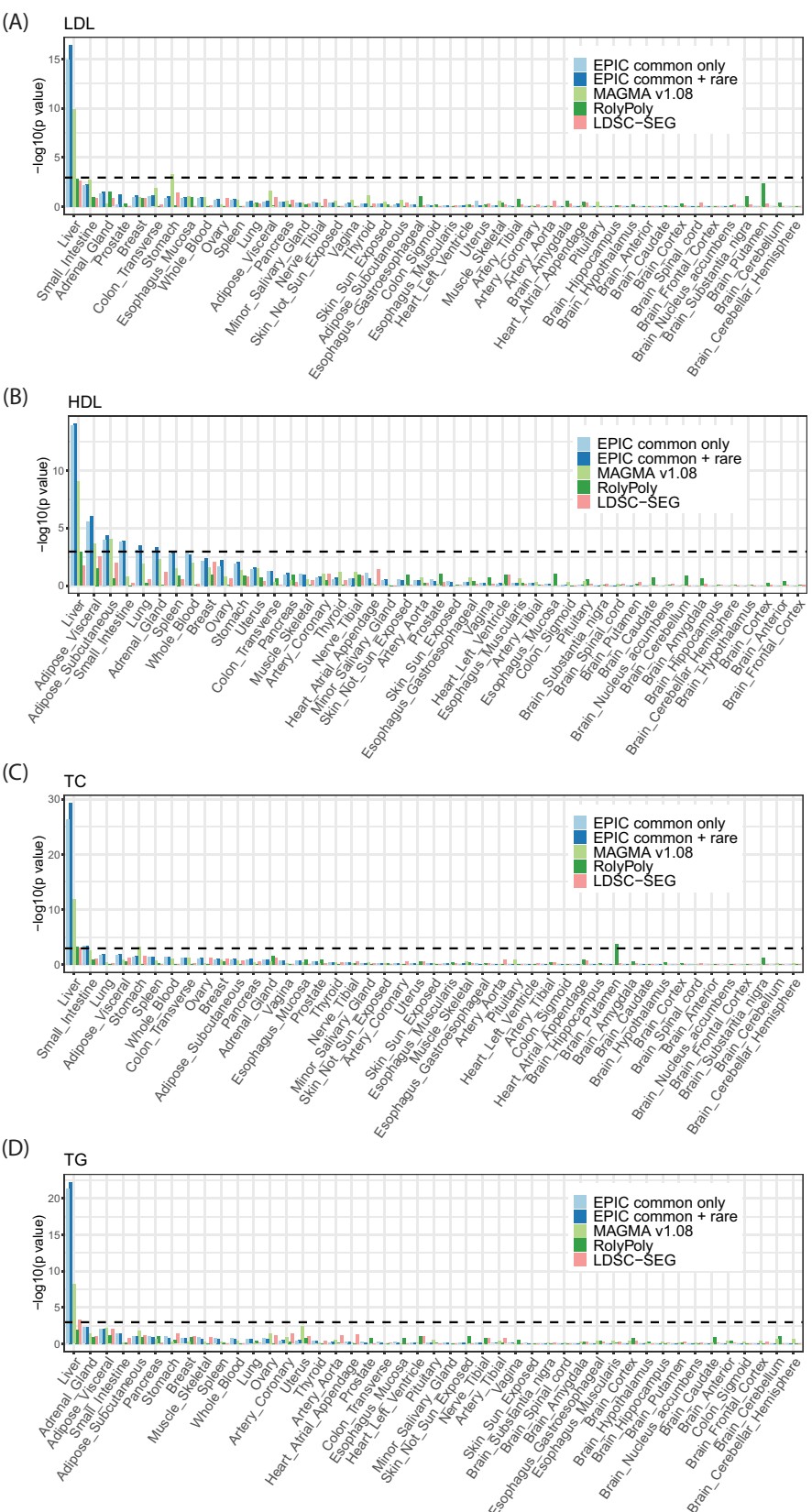

**Fig 2. Tissue enrichment for four lipid traits using GTEx bulk RNA-seq data.** (A) LDL; (B) HDL; (C) TC; and (D) TG. The dashed line is the Bonferroni-corrected *p*-value threshold. EPIC achieved higher power while controlling for false positives compared to other existing methods.

We have thus far focused on carrying out the enrichment analysis using common variants only or using common and rare variants combined. Notably, validation results based on read data suggest that combining common and rare variants is mostly better than, or otherwise on par with only using common variants. For rare variants alone, EPIC successfully identified liver and brain as the top-rank and significant tissues for the lipid traits and the neuropsychiatric disorders, respectively (Table D in S1 Text), although it is generally underpowered. This is possibly due to GWAS being underpowered to detect rare-variant associations and the expression profiles of rare variants being hard to be recapitulated by a single-cell reference. Nevertheless, while current studies only reported common variants that were consistently mapped to a subset of brain cell types for neuropsychiatric disorders [3,16], EPIC offers a statistical framework to identify cell-type-specific enrichment signals attributed to both common and rare variants, separately and jointly.

For validation, we adopted a similar strategy as proposed by Shang et al. [12]–we carried out a PubMed search, resorting to previous literature studying the trait of interest in relation to a particular tissue or cell type. Specifically, we counted the number of previous publications using the keyword pairs of trait and tissue/cell type and calculated the correlations between the number of publications and EPIC's tissue-/cell-type-specific *p*-values after negative log transformation (Fig 5). Across all traits, we found significant positive correlations between EPIC's enrichment results and PubMed search results (Fig 5A).

## Inferring relevant cell types for T2Db by scRNA-seq data of pancreatic islets

We next analyzed pancreatic islet scRNA-seq data to identify trait-relevant cell types for T2Db. To assess reproducibility, EPIC was separately applied to two scRNA-seq datasets consisting of multiple endocrine cell types (Table A and Fig D in S1 Text). The scRNA-seq data were generated using two different protocols: the SMART-Seq2 protocol on six healthy donors from Segerstolpe et al. [35] and the InDrop protocol on three healthy individuals from Baron et al. [36]. In both datasets, beta cells were identified as the trait-relevant cell types by EPIC (Fig 3B). Enrichment of beta cells is used as a gold standard for benchmark, in that beta cells produce and release insulin but are dysfunctional and gradually lost in T2Db [37]. We also found that gamma cells were marginally associated with T2Db in the Segerstolpe dataset–pancreatic polypeptide, which is produced by gamma cells, is known to play a critical role in endocrine pancreatic secretion regulation [38]. As a comparison, neither MAGMA nor LDSC-SEG detected significant enrichment in beta cells, even though the enrichment was top-ranked. RolyPoly, on the other hand, did not report any enrichment of the beta cells compared to the other types of cells.

To additionally validate the beta-cell enrichment, we carried out the PubMed search and showed that EPIC's cell-type-specific *p*-values were significantly correlated with the number of PubMed search results using the trait-and-cell-type pairs as keywords (Fig 5B). Together, we demonstrate the effectiveness of EPIC in identifying trait-relevant cell types using scRNA-seq datasets generated by different protocols.

To identify specific genes that drive the significant enrichment in beta cells, we carried out the gene-specific influence test as outlined in Methods and identified 142 highly influential genes (Fig 3C). We performed KEGG pathway analysis and GO biological process enrichment

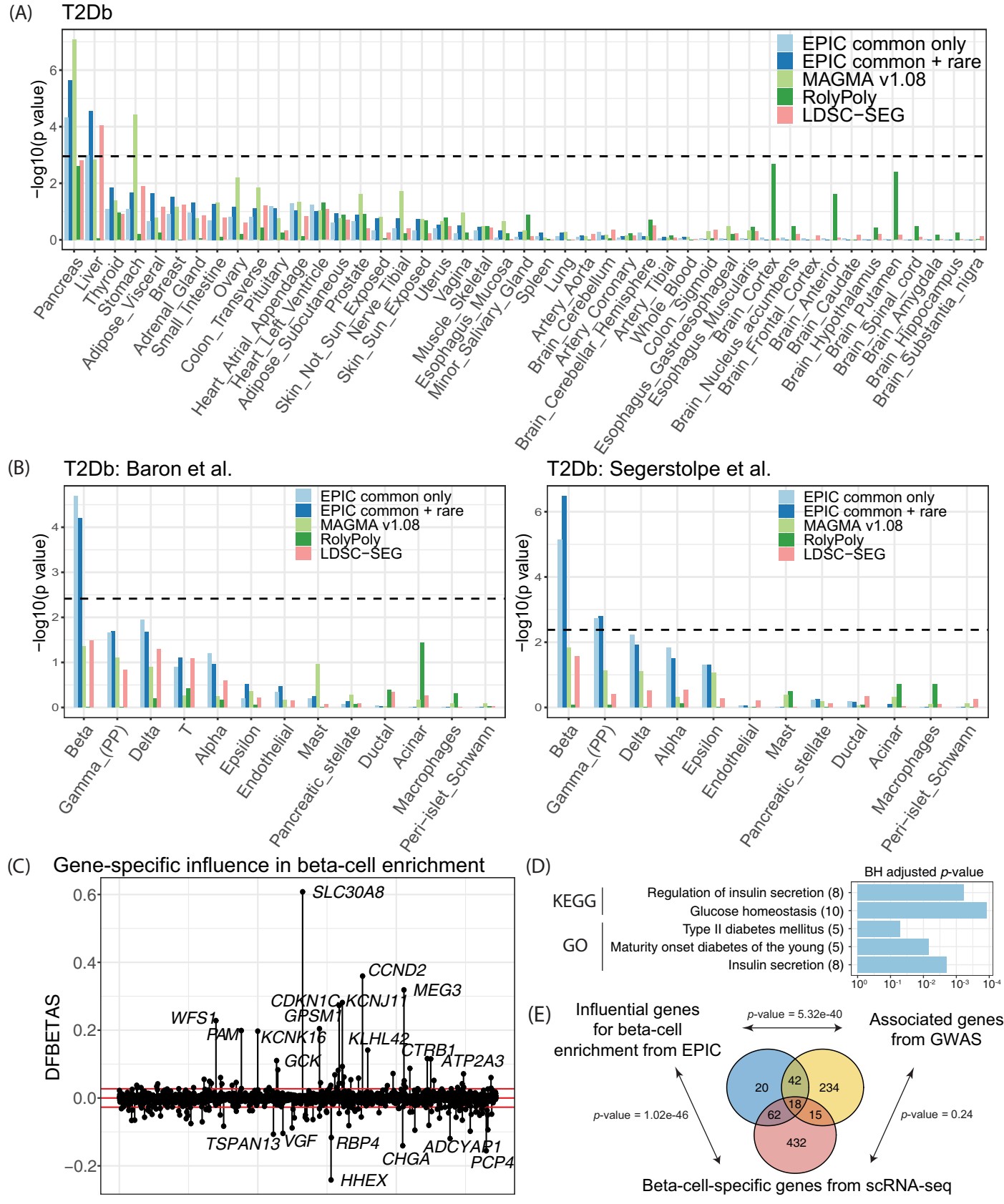

**Fig 3. Cell-type-specific enrichment of T2Db risk loci.** (A) Inferring T2Db-relevant tissues using GTEx tissue-specific RNA-seq data. (B) Inferring T2Db-relevant cell types using scRNA-seq data of human pancreatic islets. The dashed line is the Bonferroni-corrected *p*-value threshold. (C) Gene-specific influence analysis for the significantly enriched beta cells. DFBETAS measure the difference in the estimated coefficients in the gene-property analysis with and without each gene. Red lines are the size-adjusted cutoffs $\pm 2/\sqrt{N} \approx \pm 0.03$, where $N$ is the number of genes. (D) Gene pathway enrichment analysis using highly influential genes. KEGG pathways and GO biological processes related to T2Db are significantly enriched. (E). Venn diagram of the significant genes from the beta-cell-specific influential analysis by EPIC, gene-level association testing from GWAS, and nonparametric testing of cell-type-specific expression from scRNA-seq. The highly influential genes significantly overlap with genes associated with the trait and/or specifically expressed in the cell type of interest, with p-values shown from the hypergeometric test of enrichment.

analysis using the DAVID bioinformatics resources [30]. Beta-cell-specific influential genes are enriched in GO terms including glucose homeostasis and regulation of insulin secretion, as well as KEGG pathways including insulin secretion, maturity onset diabetes, etc. (Fig 3D). Compared to the set of genes that are significantly associated with T2Db from GWAS and the set of genes that are specifically expressed in beta cells from scRNA-seq, the set of

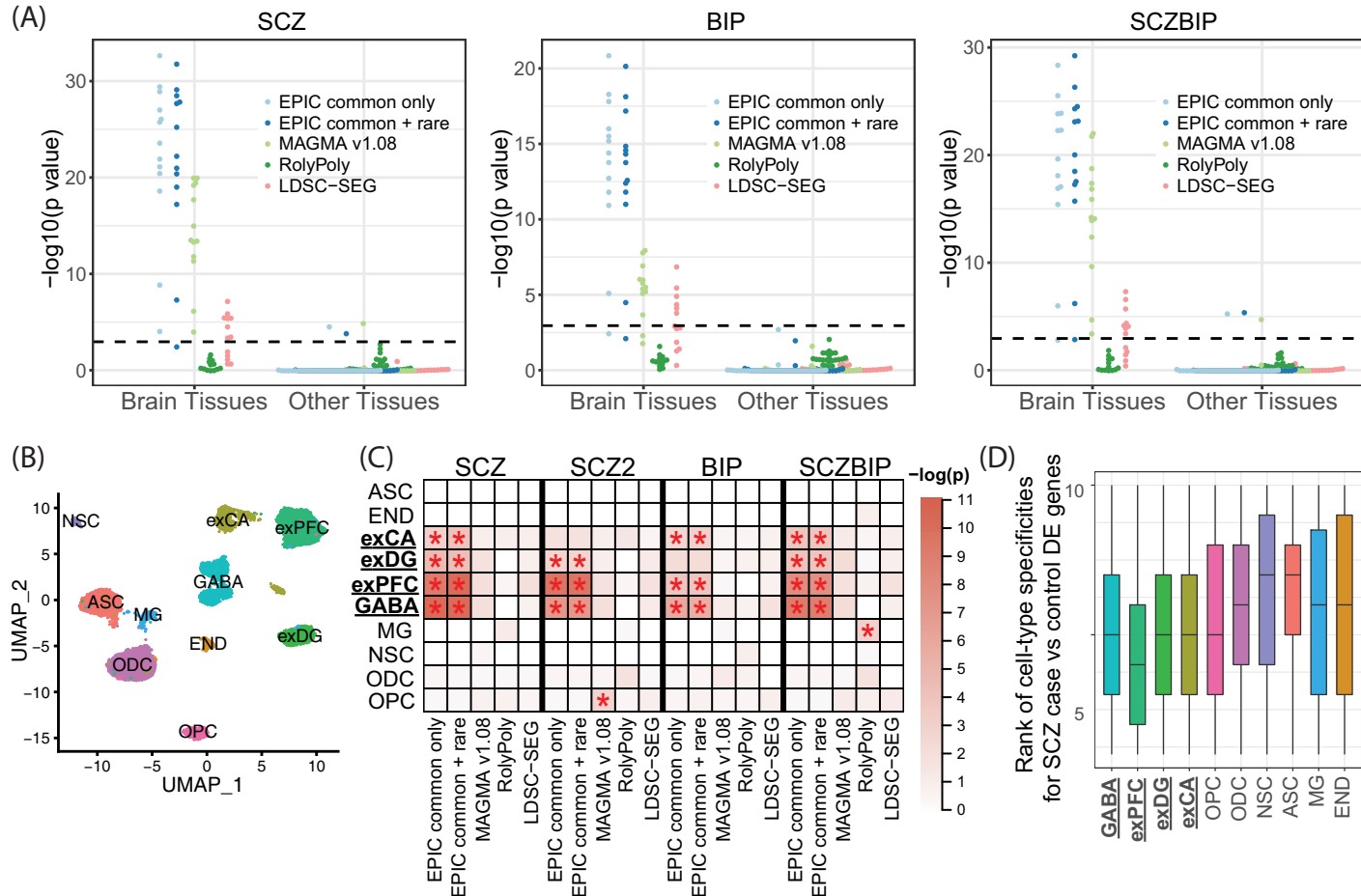

**Fig 4. Cell-type-specific enrichment for three neuropsychiatric disorders.** (A) Beeswarm plot of–log10(*p*-value) from the tissue enrichment analysis using GTEx bulk RNA-seq data. The dashed line is Bonferroni corrected *p*-value threshold. (B) UMAP embedding of 14,137 single cells from five donors. (C) Heatmap of–log10 (*p*-value) from the cell-type enrichment analysis using GTEx scRNA-seq brain data. Bonferroni-significant results are marked with red asterisks. GABA: GABAergic interneurons; exPFC: excitatory glutamatergic neurons in the prefrontal cortex; exDG: excitatory granule neurons from the hippocampal dentate gyrus region; exCA: excitatory pyramidal neurons in the hippocampal Cornu Ammonis region; OPC: oligodendrocyte precursor cells; ODC: oligodendrocytes; NSC: neuronal stem cells; ASC: astrocytes; MG: microglia cells; END: endothelial cells. (D) Boxplot of gene-specificity ranks across ten brain cell types for differentially expressed genes from SCZ case-control studies.

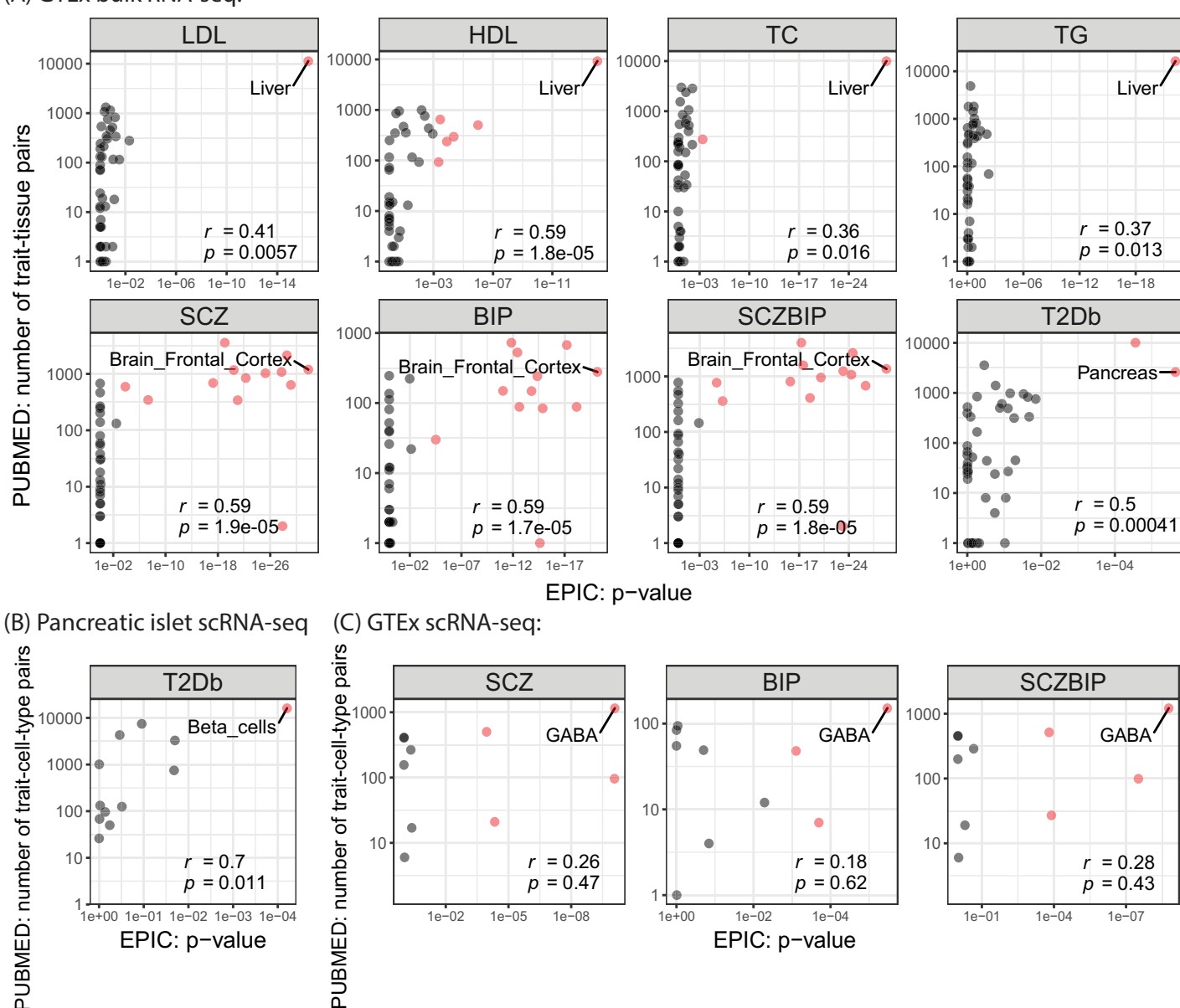

**Fig 5. Correlations of tissue/cell type ranks from enrichment analysis and PubMed Search.** Pearson correlations are calculated between the PubMed search and EPIC's tissue- or cell-type-specific enrichment testing. Trait-relevant tissues/cell types with statistical significance after Bonferroni correction are highlighted in red, where the top-ranking tissues/cell types are labeled. Correlation coefficients and *p*-values from correlation test are included.

beta-cell-specific highly influential genes identified by EPIC are highly associated with the trait and/or specifically expressed in the cell type of interest (Fig 3E). We further carried out a hypergeometric test of significant overlap between each pair of the three testing schemes (see Materials and Methods for details). Highly influential genes from EPIC's integrative analysis framework significantly overlap with genes returned by both gene-level association testing and cell-type-specific gene expression analyses, while the latter two from GWAS and scRNA-seq showed no significant overlap (Fig 3E). The influential analysis by EPIC helps prioritize trait-relevant genes in a cell-type-specific manner.

## Inferring relevant cell types for neuropsychiatric disorders by scRNA-seq data of human brain

To further test EPIC in a more complex tissue, we sought to prioritize trait-relevant cell types in the brain. While the brain tissues are significantly enriched using the GTEx bulk-tissue RNA-seq data (Fig 4A), the relevant cell types in the brain for neuropsychiatric disorders are not as well defined and studied. We obtained droplet-based scRNA-seq data [39], generated on frozen adult human postmortem tissues from the GTEx project (Table A in S1 Text), to infer the relevant cell types. After pre-processing and stringent quality controls, the scRNA-seq data contains gene expression profiles of 17,698 genes across 14,137 single cells collected from the human hippocampus and prefrontal cortex tissues of five donors. The cells belong to ten cell types (Fig 4B), and we focused on the top 8,000 highly variable genes for subsequent analyses.

We evaluated EPIC's cell-type-specific enrichment results and found that all three neuropsychiatric disorders were significantly enriched in GABAergic interneurons (GABA), excitatory glutamatergic neurons from the prefrontal cortex (exPFC), and excitatory pyramidal neurons in the hippocampal CA region (exCA). Additionally, excitatory granule neurons from the hippocampal dentate gyrus region (exDG) were identified as relevant cell types for SCZ and SCZBIP (Fig 4C).

We employed three strategies to validate the trait-relevant cell types for the neuropsychiatric disorders. First, we again found positive correlations between PubMed search results and EPIC's enrichment results (Fig 5C). Although the correlation testing does not retain significance, the top enriched cell type (GABA), when paired with the three psychiatric traits, also returned the largest number of PubMed hits (Fig 5C). The insignificant $p$-values from the correlation test are likely due to the limited number of existing single-cell studies on neuropsychiatric diseases via PubMed; indeed, the cell types that are found to be enriched by EPIC agree with the recently reported association of neuropsychiatric disorders with interneurons and excitatory pyramidal neurons [3, 16]. Second, we adopted additional independent GWAS summary statistics for SCZ (SCZ2) [40] (Table A in S1 Text) and observed highly concordant enrichment results between SCZ and SCZ2 (Fig 4C). Third, we tested whether genes that are upregulated/downregulated for SCZ were enriched in the identified cell types to additionally implicate cell types involved in SCZ. Specifically, we performed differential expression (DE) analysis from an independent SCZ case-control study using bulk RNA-seq of the dorsolateral prefrontal cortex [41], retaining 287 significant DE genes that also overlap the post-QC genes from scRNA-seq (Fig E in S1 Text). We reasoned that if SCZ-relevant risk loci were enriched in a particular cell type, genes that are differentially expressed between SCZ cases and controls would demonstrate greater cell-type specificity in this cell type. We calculated cell-type specificities using the set of DE genes and observed GABA, exCA, exDG, and exPFC were the top four cell types with the lowest gene-specificity ranks (Fig 4D). Using three different strategies by querying external databases and adopting additional and orthogonal datasets, we validated the trait-cell-type relevance results.

## Simulation studies to assess power and type I error control

To assess EPIC's type I error control for gene-level association testing, we first carried out simulation studies under the null, where we resampled the reference genotype matrix and generated binary phenotypes from a Bernoulli distribution with probabilities 0.1, 0.2, and 0.5. We then computed the SNP-level summary statistics using logistic regression and applied EPIC to carry out the gene-level testing under this null setting. For both large and small genes (with number of SNPs greater/less than 50), EPIC controls for false positives (Fig F in S1 Text). Refer to Materials and Methods for simulation details.

To assess EPIC's type I error control for the second-step regression analysis, we randomly permuted the gene expressions to disrupt any association between gene-level summary statistics and their expression profiles. We applied EPIC to prioritize tissues/cell types: EPIC controls false positives with type I error rates close to zero for the lipid traits, less than 0.02 for the psychiatric diseases, and less than 0.01 for the T2Db (Fig G in S1 Text).

To assess EPIC's power for gene-level association testing, we generated dichotomous phenotypes as a function of the genotypes with varied proportions of causal variants, effect sizes, and directions of effects (see Materials and Methods for details). We then computed the summary statistics for each SNP via logistic regression, which were used as input for EPIC. Our results suggest that under 30 different simulation configurations EPIC is more powerful than MAGMA and that the power gain is substantial when the signals are sparse or in different directions (Table E in S1 Text).

## Discussion

Over the last one and half decades, GWASs have successfully identified and replicated genetic variants associated with various complex traits. Meanwhile, bulk-tissue and single-cell transcriptomic sequencing allow tissue- and cell-type-specific gene expression characterization and have seen rapid technological development with ever-increasing sequencing capacities and throughputs. Here, we propose EPIC to address the problem of how GWAS summary statistics should be integrated with scRNA-seq data to prioritize trait-relevant cell type(s) and to elucidate disease etiology. To our best knowledge, EPIC is the first method that prioritizes cell type(s) for both common and rare variants with a rigorous statistical framework that properly accounts for both within- and between-gene correlations. We demonstrate EPIC's effectiveness and outperformance compared to existing methods with extensive benchmark and validation studies. Notably, EPIC's analysis can be run in parallel across different chromosomes, and its overall computational efficiency is on par with the other existing methods from benchmark analyses on identifying relevant GTEx tissues for the LDL, SCZ, and T2Db traits (Table F in S1 Text).

For scRNA-seq data, all existing methods, including EPIC, resort to pre-clustered/annotated cell types and average across cells to obtain cell-type-specific expression profiles. However, scRNA-seq goes beyond the mean measurements [42,43], and how to make the best use of gene expression dispersion, nonzero fraction, and other aspects of its distribution needs further method development [44]. Additionally, while many efforts have been devoted to identifying enrichment of discretized cell types, how to carry out enrichment analysis for transient cell states needs further investigation. Last but not least, when multiple scRNA-seq datasets are available across different experiments, protocols, or species, borrowing information from additional sources can potentially boost the performance and increase the robustness of the enrichment analysis [45]. While it is nontrivial to directly perform gene expression data integration, a cross-dataset conditional analysis workflow was proposed by Watanabe et al. [17] to evaluate the association of cell types based on multiple independent scRNA-seq datasets. However, the linear conditional analysis may not be sufficient to capture any nonlinear batch effects [46,47].

It is also worth noting that CoCoNet, MAGMA, and EPIC first carry out a gene-level association test so that the summary statistics and expressions are unified to be gene-specific. They integrate SNP-wise summary statistics in different ways, yet for all methods, SNPs need to be first annotated to genes based on a window surrounding each gene. While RolyPoly and LDSC-SEG model on the SNP level directly, each SNP still needs to be assigned to a gene so that the gene expression can be used as an SNP annotation. There is not a consensus on how

to most accurately assign SNPs to genes, and more importantly, one would only be able to perform gene annotations for SNPs that reside in gene bodies or promoter regions. Meanwhile, a large number of GWAS hits are in the noncoding regions, and their functions are yet to be fully understood. EPIC's framework can be easily extended to infer enrichment of noncoding variants when combined with the single-cell assay for transposase-accessible chromatin using sequencing (ATAC-seq) data [48,49]. Additionally, cell-type-specific expression quantitative trait loci from the noncoding regions [50] can also be integrated with the second-step gene-property analysis to boost power and to infer enrichment of noncoding variants.

## Materials and methods

### Gene-level associations for common variants

Let $\beta = (\beta_1, \cdots, \beta_k)^T$ be the effect sizes of $K$ common variants within a gene of interest. Let $\hat{\beta} = (\hat{\beta}_1, \cdots, \hat{\beta}_K)^T$ be the estimator for $\beta$, with corresponding standard error $\hat{\sigma} = (\hat{\sigma}_1, \cdots, \hat{\sigma}_K)^T$. Let $\hat{z} = (\hat{z}_1, \cdots, \hat{z}_K)^T$ be the $z$-scores, where $\hat{z}_j = \hat{\beta}_j / \hat{\sigma}_j$ is the standard-normal statistic for testing the null hypothesis of no association for SNP $j$. We approximate the correlation matrix of $\hat{\beta}$ (equivalent to the covariance matrix of $\hat{z}$) by the LD matrix $R = \{R_{jl}; j, l = 1, \ldots, K\}$, where $R_{jl}$ is the Pearson correlation between SNP $j$ and SNP $l$. We further define $V = \mathrm{cov}(\hat{\beta}) = \mathrm{diag}(\hat{\sigma}) \, R \, \mathrm{diag}(\hat{\sigma})$ as the covariance matrix of $\hat{\beta}$. Under the null hypothesis of $\beta = 0$, the estimator $\hat{\beta}$ is $K$-variate normal with mean 0 and covariance matrix $V$. To perform gene-level association testing for common variants, we construct a simple and powerful chi-square statistic for testing the null hypothesis of $\beta = 0$:

$$Q^c = \hat{\beta}^T V^{-1} \hat{\beta} = \hat{z}^T R^{-1} \hat{z},$$

which has the $\chi_K^2$ distribution under the null. The correlation matrix $R$ can be estimated from either the raw genotype matrix or a publicly available reference panel. In this study, we utilize the 1000 Genomes Project European panel [51], which comprises genotypes of ~500 European individuals across ~23 million SNPs.

An effective chi-square test described above requires the covariance matrix to be well-conditioned. For most GWASs, the ratio of the number of SNPs and the number of subjects is greater than or close to one, making the sample covariance matrix ill-conditioned [52]. In these cases, smaller eigenvalues of the sample covariance matrix are underestimated, leading to inflated false positives in the gene-level association testing. To solve this issue, we choose to adopt the POET estimator [53], a principal orthogonal complement thresholding approach, to obtain a well-conditioned covariance matrix via sparse shrinkage under a high-dimensional setting. The estimator of $V = \{V_{jl}; j, l = 1, \ldots, K\}$ is defined as $\hat{V}_H = \sum_{j=1}^H \hat{\lambda}_j \hat{v}_j \hat{v}_j^T + \hat{R}_H^*$, where $\hat{\lambda}_j$ is the $j$th eigenvalues of the covariance matrix with corresponding eigenvector $\hat{v}_j$, $\hat{R}_H^*$ is obtained from applying adaptive thresholding on $\hat{R}_H = \sum_{j=H+1}^K \hat{\lambda}_j \hat{v}_j \hat{v}_j^T$, and $H$ is the number of spiked eigenvalues. The degree of shrinkage is determined by a tuning parameter, and we choose one so that the positive definiteness of the estimated sparse covariance matrix is guaranteed. Notably, other sparse covariance matrix estimators [52,54,55] can also be used in a similar fashion.

### Gene-level associations for rare variants

Recent advances in next-generation sequencing technology have made it possible to extend association testing to rare variants, which can explain additional disease risk or trait variability

[26,28,56]. Previous work [57] has demonstrated that gene-based association tests for rare variants can be constructed using the single-variant statistics and that it is powerful and achieves well-controlled type I error as long as the correlation matrix of single-variant test statistics can be accurately estimated. Here, we inherit the same framework and recover the gene-level burden test statistics [58,59] for rare variants from univariate GWAS summary statistics. Specifically, let $U = \{U_j; j = 1,\ldots,M\}$ and $C = \{C_{jl}; j,l = 1,\ldots,M\}$ be the score statistic and the corresponding covariance matrix for testing the null hypothesis of no association for a total of $M$ rare variants residing in a gene. Under $H_0$, the burden test statistic $T = \xi^T U/\sqrt{\xi^T C \xi}$ follows a standard normal distribution, where $\xi_{M\times 1} = (1,\cdots,1)^T$. The GWAS summary statistics do not contain $U$ and $C$; we approximate $U_j$ and $C_{jl}$ by

$$\hat{U}_j = w_j \hat{\beta}_j / \hat{\sigma}_j = w_j \hat{z}_j$$

$$\hat{C}_{jl} = w_j R_{jl} w_l,$$

where $R$ is the correlation or covariance matrix of $\hat{z}$ and $w_j = 1/\hat{\sigma}_j$ is an empirical approximation to $\sqrt{C_{jj}}$. Denote $w = (w_1,\ldots,w_M)^T$. The burden test uses $Q^r = (w^T \hat{z})^2/w^T R w$, which follows the $\chi_1^2$ distribution under the null.

## Joint analysis for common and rare variants

Existing methods either remove rare variants from the analysis [9,11] or do not differentiate common and rare variants when only summary statistics are available [10]. Yet, existing GWASs have successfully uncovered both common and rare variants associated with complex traits and diseases [26,28,56], and rare variants should therefore not be ignored in the enrichment analysis. To incorporate rare variants into the common-variant gene association testing framework, we collapse genotypes of all rare variants within a gene to construct a pseudo-SNP. We then treat the aggregated pseudo-SNP as a common variant and concatenate the $z$-scores $\hat{z}^* = (\hat{z}_1,\cdots,\hat{z}_K,\hat{z}^r)^T$, where the first $K$ elements are from the common variants and $\hat{z}^r = w^T \hat{z}/\sqrt{w^T R w}$ is from the burden test statistic for the combined rare variants. A joint chi-square test for common and rare variants is performed as below:

$$Q = \hat{z}^{*T} R^{*-1} \hat{z}^*,$$

which has the $\chi_{K+1}^2$ distribution under the null hypothesis. $R^*$ can be estimated using POET shrinkage with the pseudo-SNP included.

## Gene-gene correlation

Proximal genes that share *cis*-SNPs inherit LD from SNPs and result in correlations among genes. Since the correlations between genes are caused by LD between SNPs, which quickly drops off as a function of distance, we adopt a sliding-window approach to only compute correlations for pairs of genes within a certain distance from each other. It is worth noting that this also significantly reduces the computational burden. Specifically, let $N$ be the number of genes from the same chromosome, and we adopt a sliding window of size $d$ to estimate the sparse covariance matrix among genes $\{G_1,\ldots,G_d\}, \{G_2,\ldots,G_{d+1}\},\ldots,\{G_{N-d+1},\ldots,G_N\}$, respectively. By default, we set $d = 10$ so that gene-wise correlations can be recovered for a gene with its 18 neighboring genes (see Fig H in S1 Text for the effect of sliding window size on EPIC's performance). Similar to MAGMA, correlations are only computed for pairs of genes within 5 megabases by default.

Recall that the gene-level association statistics are chi-square statistics in a quadratic form. Within a specific window, the gene-wise correlations are obtained via transformations of the SNP-wise LD information. Let $\hat{z}^{(s)}$ and $\hat{z}^{(t)}$ be the SNP-wise $z$-scores for genes $s$ and $t$, respectively. Let $R^{(s)} = \{R_{jl}^{(s)}; j, l = 1, \ldots, K_s\}$, $R^{(t)} = \{R_{jl}^{(t)}; j, l = 1, \ldots, K_t\}$, and $R^{(s,t)} = \{R_{jl}^{(s,t)}; j = 1, \ldots, K_s, l = 1, \ldots, K_t\} = \mathrm{cor}(\hat{z}^{(s)}, \hat{z}^{(t)})$ be the within- and between-gene correlation matrices obtained from the POET shrinkage estimation. We take advantage of the Cholesky decomposition to obtain the gene-gene correlation between $Q_s = (\hat{z}^{(s)})^T (R^{(s)})^{-1} \hat{z}^{(s)}$ and $Q_t = (\hat{z}^{(t)})^T (R^{(t)})^{-1} \hat{z}^{(t)}$:

$$\rho_{st} = \mathrm{cor}(Q_s, Q_t) = \frac{\sum_{j=1}^{K_s} \sum_{i=1}^{K_s+K_t} L_{ij}^2}{\sqrt{K_s K_t}},$$

where $L_{ij}$'s are entries of a lower triangular matrix $L$ such that $\tilde{R}_{(K_s+K_t) \times (K_s+K_t)} = LL^T$ and

$$\tilde{R}_{(K_s+K_t) \times (K_s+K_t)} = \begin{pmatrix} I_{K_s} & R^{(s)^{-1/2}} R^{(s,t)} R^{(t)^{-1/2}} \\ R^{(t)^{-1/2}} R^{(t,s)} R^{(s)^{-1/2}} & I_{K_t} \end{pmatrix},$$

$I_K$ is the identity matrix with dimension $K$. The full derivation is detailed in Note B in S1 Text. When rare variants are included in the framework, gene-gene correlations are calculated similarly by aggregating all rare variants that reside in a gene as a pseudo-SNP.

## Prioritizing trait-relevant cell type(s)

To identify cell-type-specific enrichment for a specific trait of interest, we devise a regression framework based on generalized least squares to identify risk loci enrichment. The key underlying hypothesis is that if a particular cell type influences a trait, more GWAS polygenic signals would be concentrated in genes with greater cell-type-specific gene expression. Under this hypothesis, genes that are significantly associated with lipid traits are expected to be highly expressed in the liver since the liver is known to participate in cholesterol regulation. This relationship between the GWAS association signals and the gene expression specificity is modeled as below.

Let $Q_g$ be the gene-level chi-square association test statistic for gene $g$. To account for the different number of SNPs within each gene, we adjust the degree of freedom of $K_g+1$ to obtain $Y_g = Q_g/(K_g+1)$, which is included as the outcome variable. Note that under the null, $Y_g$ has mean of 1 and variance of $2/(K_g+1)$. For each cell type $c$, to test for its enrichment we fit a separate regression using its cell-type-specific gene expression $E_{cg}$ (reads per kilobase million (RPKM) or transcripts per million (TPM)) as a dependent variable. To account for the baseline gene expression [17], we also include another covariate $A_g = \frac{1}{T} \sum_{c=1}^{T} E_{cg}$, which is the average gene expression across all $T$ cell types. Taken together, we have

$$Y = \gamma_0 + E_c \gamma_c + A \gamma_A + \epsilon,$$

where $\epsilon$ has a multivariate normal distribution with mean 0 and covariance $\sigma^2 W$, $W = DPD^T$, $D = \mathrm{Diag}(\sqrt{2/(K_g + 1)})$, and $P = \{\rho_{st}\}$ is the gene-gene correlation matrix of the chi-square statistics. We adopt the generalized least squares approach to fit the model and perform a one-sided test against the alternative $\gamma_c > 0$, under which the gene-level association signals positively correlated with the cell-type-specific expression.

For a significantly enriched cell type, we further carry out a statistical influence test to identify a set of cell-type-specific influential genes, using the DFBETAS statistics, which indicate the effect that deleting each observation has on the estimates for the regression coefficients

[60]. Specifically, we repeatedly remove gene $g$, fit the regression model, and denote the estimated coefficient as $\gamma_c^{(g)}$. DFBETA$_g \equiv \gamma_c - \gamma_c^{(g)}$, and large values of DFBETAS indicate observations (i.e., genes) that are influential in estimating $\gamma_c$. With a size-adjusted cutoff $2/\sqrt{N}$, where $N$ is the number of genes used in the cell-type-specific enrichment analysis, significantly influential genes allow for further pathway or gene set enrichment analyses.

To compare these influential genes from EPIC's integrative framework with significant genes from gene-level association testing (by GWAS) and cell-type-specific gene expression analyses (by scRNA-seq), we carry out a hypergeometric test of significant overlap between each pair of the three call sets. Specifically, let $m$ and $n$ be the number of significant and insignificant genes for testing A, respectively, $k$ be the number of significant genes from testing B, and $q$ be the number of overlapped significant genes between A and B. We aim to test for an enrichment of testing A's significant genes in B–the $p$-value of enrichment is derived from the hypergeometric distribution using the cumulative distribution function, coded as phyper(q, m, n, k, lower.tail = FALSE) in R.

## GWAS summary statistics and transcriptomic data processing

We adopt GWAS summary statistics of eight traits, including four lipid traits [24] (low-density lipoprotein cholesterol (LDL), high-density lipoprotein cholesterol (HDL), total cholesterol (TC), and triglyceride levels (TG)), three neuropsychiatric disorders [25–27] (schizophrenia (SCZ), bipolar disorder (BIP), and schizophrenia and bipolar disorder (SCZBIP)), and type 2 diabetes [28] (T2Db). The relevant tissues involved in these traits are well known/studied–liver for the lipid traits, brain for the neuropsychiatric disorders, and pancreas for the T2Db–and we use this as ground truths to demonstrate EPIC and to benchmark against other methods. See Table A in S1 Text for more information on the GWASs.

For each trait, we obtain SNP-level summary statistics and apply stringent quality control procedures to the data. We restrict our analyses to autosomes, filter out SNPs not in the 1000 Genomes Project Phase 3 reference panel, and remove SNPs with mismatched reference SNP ID numbers. We exclude SNPs from the major histocompatibility complex (MHC) region due to complex LD architecture [9,12,14]. In addition to SNP filtering, we align alleles of each SNP against those of the reference panel to harmonize the effect alleles of all processed GWAS summary statistics. A gene window is defined with 10kb upstream and 1.5kb downstream of each gene [3], and SNPs residing in the windows are assigned to the corresponding genes.

In the analysis that follows, we uniformly report results using a minor allele frequency (MAF) cutoff of 1% to define common and rare variants (see Fig I in S1 Text for enrichment results with different MAF cutoffs). To reduce the computational cost and to alleviate the multicollinearity problem, we perform LD pruning using PLINK [61] with a threshold of $r^2 \leq 0.8$ to obtain a set of pruned-in common variants, followed by a second-round of LD pruning if the number of common SNPs per gene exceeds 200. See Fig J in S1 Text for results with varying LD-pruning thresholds. For rare variants, we only carry out a gene-level rare variant association testing if the minor allele count (MAC), defined as the total number of minor alleles across subjects and SNPs within the gene, exceeds 20. We report the number of SNPs (common variants and rare variants), the number of genes, and the number of SNPs per gene for each GWAS trait in Table G in S1 Text.

We adopt a unified framework to process all transcriptomic data. For scRNA-seq data, we follow the Seurat [47] pipeline to perform gene- and cell-wise quality controls and focus on the top 8000 highly variable genes. Cell-type-specific RPKMs are calculated by combining read or UMI counts from all cells of a specific cell type, followed by log2 transformation with an added pseudo-count. For tissue-specific bulk RNA-seq data from GTEx, we first calculate a tissue

specificity score for each gene [12], and only focus on genes that are highly specific in at least one tissue. See Note A in S1 Text for more details. We then perform log2 transformation on the tissue-specific TPM measurements with an added pseudo-count.

## Benchmarking against RolyPoly, LDSC-SEG, and MAGMA

We benchmarked EPIC against three existing approaches: RolyPoly [9], LDSC-SEG [11], and MAGMA [10]. For all methods, we used RPKMs for each cell type and TPMs for each GTEx tissue in the benchmarking analysis. We made gene annotations the same for RolyPoly, MAGMA, and EPIC by defining the gene window as 10kb upstream and 1.5kb downstream of each gene. For LDSC-SEG [11], as recommended by the authors, the window size is set to be 100kb up and downstream of each gene's transcribed region. Since all methods adopt a hypothesis testing framework to identify trait-relevant cell type(s), for each pair of trait and cell type, we reported and compared the corresponding $p$-values from the different methods.

RolyPoly takes as input GWAS summary statistics, gene expression data, gene annotations, and LD matrix from the 1000 Genomes Project Phase 3. As recommended by the developer for RolyPoly [9], we scaled the gene expression for each gene across cell types and took the absolute values of the scaled expression values. We performed 100 block bootstrapping iterations to test whether a cell-type-specific gene expression annotation was significantly enriched in a joint model across all cell types. We also benchmarked LDSC-SEG, which computes $t$-statistics to quantify differential expression for each gene across cell types. We annotated genome-wide SNPs using the top 10% genes with the highest positive $t$-statistics and applied stratified LDSC to test the heritability enrichment of the annotations that were attributed to specifically expressed genes for each cell type. For MAGMA, we first obtained gene-level association statistics using MAGMA v1.08. We then carried out the gene-property analysis proposed in Watanabe et al. [17], with technical confounders being controlled by default, to test the positive relationship between cell-type specificity of gene expression and genetic associations.

## Simulation setup

We first ran a null simulation to evaluate EPIC's type I error control for its gene-level association testing. We generated simulated null genes in the following way. (i) We randomly selected 100 genes with well-pruned and annotated common SNPs from our SCZ GWAS analysis; the number of SNPs per gene ranged from 10 to 195, with a mean of 53.2 and a median of 31.5. (ii) For each gene, we resampled the reference genotype matrix from the 1000 Genomes Project 5,000 times and obtained a resampled reference genotype matrix $X_{n \times m}$, where $n = 5000$ individuals, $m$ = the number of SNPs, and $X_{ij} \in \{0,1,2\}$ indicates the number of effect alleles for individual $i$ and SNP $j$. (iii) We simulated binary phenotype $Y_i \sim \text{Bernoulli}(\theta)$ for each individual $i$ and $\theta \in \{0.1, 0.2, 0.5\}$. For each SNP $j$, we fit a logistic regression to generate SNP-level $z$-scores with corresponding two-sided $p$-values. (iv) To compute the correlation matrix for each gene, we obtained the sparse and shrinkage POET estimators within the sliding window. (v) Repeat (ii)—(iv) 1,000 times and calculate the type I error rate for small genes (number of SNPs $\leq 50$) and large genes (number of SNPs $> 50$).

To evaluate EPIC's power for gene-level association testing, we used the same set of 100 genes with the same resampled reference genotype matrix as mentioned above. For each gene, we randomly selected different proportions of causal variants with different effect sizes and generated dichotomous phenotypes under the alternative by

$$\text{logit}P(Y_i = 1) = \sum_j \beta_j X_{ij}.$$

Balanced case-control sampling was used to simulate dichotomous phenotypes with 50% cases and 50% controls [29,62]. For each SNP, we then fit a logistic regression to generate SNP-level $z$-scores with corresponding two-sided $p$-values as summary statistics. The proportion of causal variants was set to be 5%, 20%, and 50%, representative of both sparse and dense signals; effect sizes, as well as directions of effects, were also varied (Table E in S1 Text). Altogether, we had a total of 30 simulation configurations for each of the 100 genes. Empirical power was estimated as the proportion of $p$-values less than $\alpha = 10^{-6}$. The simulation was repeated 1,000 times to allow for standard error estimates.

## Supporting information

**S1 Text.** Supplementary materials including Note A-B, Table A-G, and Figure A-J. (DOCX)

## Acknowledgments

The authors thank Drs. Yun Li, Michael Love, Karen Mohlke, and Jason Stein for helpful discussions and comments, and Drs. Alkes Price, Diego Calderon, and Kyoko Watanabe for providing support and insight on existing methods.

## Author Contributions

**Conceptualization:** Dan-Yu Lin, Yuchao Jiang.

**Data curation:** Rujin Wang.

**Formal analysis:** Rujin Wang, Dan-Yu Lin, Yuchao Jiang.

**Funding acquisition:** Dan-Yu Lin, Yuchao Jiang.

**Investigation:** Rujin Wang, Dan-Yu Lin, Yuchao Jiang.

**Methodology:** Rujin Wang, Dan-Yu Lin, Yuchao Jiang.

**Project administration:** Dan-Yu Lin, Yuchao Jiang.

**Resources:** Rujin Wang, Dan-Yu Lin, Yuchao Jiang.

**Software:** Rujin Wang.

**Supervision:** Dan-Yu Lin, Yuchao Jiang.

**Validation:** Rujin Wang, Yuchao Jiang.

**Visualization:** Rujin Wang, Yuchao Jiang.

**Writing – original draft:** Rujin Wang, Yuchao Jiang.

**Writing – review & editing:** Rujin Wang, Dan-Yu Lin, Yuchao Jiang.

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
