## [Decision Letter · Decision Letter 0]

2 Feb 2022

Dear Dr Jiang,

Thank you very much for submitting your Research Article entitled 'EPIC: inferring relevant cell types for complex traits by integrating genome-wide association studies and single-cell RNA sequencing' to PLOS Genetics.

The manuscript was fully evaluated at the editorial level and by independent peer reviewers. The reviewers appreciated the attention to an important problem, but raised some substantial concerns about the current manuscript. Based on the reviews, we will not be able to accept this version of the manuscript, but we would be willing to review a much-revised version. We cannot, of course, promise publication at that time.

Both reviewer 2 and 3 have indicated that simulations should be done to understand why EPIC is more powerful than the existing methods. We concur with the reviewers and therefore strongly suggest the authors to provide additional simulation results. 

If you decide to revise the manuscript for further consideration at PLOS Genetics, please aim to resubmit within the next 60 days, unless it will take extra time to address the concerns of the reviewers, in which case we would appreciate an expected resubmission date by email to plosgenetics@plos.org.

[LINK]

We are sorry that we cannot be more positive about your manuscript at this stage. Please do not hesitate to contact us if you have any concerns or questions.

Yours sincerely,

Xiaofeng Zhu

Associate Editor

PLOS Genetics

David Balding

Section Editor: Methods

PLOS Genetics

**Comments to the Authors:**

Reviewer #1: Genome-wide association studies (GWASs) have yielded genetic variants associated with various complex traits. Emerging evidence suggests that the function of trait associated variants likely acts in a tissue- or cell-type-specific fashion. For many complex traits, however, the specific cell or tissue types leading to risk are unknown. Recent advances of single-cell RNA sequencing (scRNA-seq) provide unprecedented opportunities, alongside challenges, to systematically investigate the cell-type-specific enrichment of GWAS risk variants. Authors propose EPIC, a statistical framework that relates large-scale GWAS summary statistics to cell-type-specific transcriptomic measurements from scRNA-seq data to prioritize trait-relevant cell types. They use known trait-relevant tissues and cell types as ground truths for benchmark, adopt independent GWAS and scRNA-seq datasets for reproducibility, and refer to PubMed keyword search and existing case-control studies for validation. Such an integrative analysis helps elucidate the underlying cell-type-specific disease etiology and prioritize important risk variants. I have following comments.

1. The idea of this manuscript is not novel. It is not the first that relates GWAS results with scRNA-seq datasets. In Watanabe et al. [1], they have already proposed almost the same framework to connect GWAS signals with cell-type-specific from scRNA-seq datsets, where they first converted SNP-based summary statistics into gene-based ones using MAGMA and using the processed single-cell cell-type-specific transcriptome datasets to associate with gene-based Z-scores in a fashion similar to the one on line 436. It is not clear what the major difference between these two methods. Authors should clearly state and discuss it.

2. In my view, the contribution of this manuscript is that they derive a gene-based test for common and rare variants. I have few questions regarding this. First, how LD can be estimated using reference panel for rare variants? The pseudo-SNP strategy depends on the Burden test, where it assumes that all variants have the same direction of effects. The authors need to show the effectiveness of using this as surrogate. Second, genes are not distributed equally across the genome. Some regions are denser and some are sparser. How do we know that a sliding window of size d will capture the major correlated genes?

3. In the main results, it looks like that at least four out of eight traits (Figure S1) have problem of inflated p-values, especially for SCZ and T2D. Before making any statement of more significant findings, one need to demonstrate its type I error control. Authors only use housekeeping genes to show this point but I cannot find the nominal p-value or qq-plot. It is insufficient to claim that the method achieves well-controlled type I error.

[1] Watanabe, K., Mirkov, M. U., de Leeuw, C. A., van den Heuvel, M. P., & Posthuma, D. (2019). Genetic mapping of cell type specificity for complex traits. Nature communications, 10(1), 1-13.

Reviewer #2: Wang et al. presented a novel method EPIC that relates large-scale GWAS summary statistics to cell-type-specific gene expression measurements from single-cell RNA sequencing. The novel features of EPIC include a powerful gene-level test statistics for both common and rare variants, and an adaptation of generalized least squares to prioritize trait relevant cell types while accounting for the correlation structures both within and between genes. EPIC is applied to multiple scRNA-seq datasets from different platforms and identify cell types underlying type 2 diabetes and schizophrenia and outperforms existing approaches include LDSC-SEG and RolyPoly. Overall, this is a very well written paper with a nice method. I enjoyed reading the paper and only have a few comments that hopefully will help further improve the quality of the paper:

There are two major advantages of EPIC: (1) it models the correlation structures both within and between genes (while LDSC-SEG and RolyPoly directly model correlation among SNPs), and (2) it uses a new framework to link the gene-level z-scores with gene expression level (while LDSC-SEG and RolyPoly uses evidence for differential gene expression instead of expression level directly). It would be helpful to run simulations to illustrate which advantage explains the majority of EPIC's power gain over LDSC-SEG and RolyPoly. Would it also be possible to disentangle these two advantages in the real data applications?

The gene-level association test reminds me of the variance component model used in SKAT. How does the association test in EPIC compare to SKAT, or a similar variance component test that incorporates common variants only, or both common and rare variants?

For Figure S3, could you show a qq-plot instead? The figure legend is unclear, and I couldn't figure out how many genes are included in these boxplots (52, 2088, or 8708?). Similarly, for Figure S4, could you show a qq-plot to see whether p-values are calibrated across a range of cutoffs?

Reviewer #3: The review is uploaded as an attachment.

**Have all data underlying the figures and results presented in the manuscript been provided?**

Reviewer #1: Yes

Reviewer #2: Yes

Reviewer #3: Yes

PLOS authors have the option to publish the peer review history of their article (what does this mean?). If published, this will include your full peer review and any attached files.

Reviewer #1: No

Reviewer #2: No

Reviewer #3: **Yes: **Arunabha Majumdar

---

## [Decision Letter · Decision Letter 1]

10 May 2022

Dear Dr Jiang,

Thank you very much for submitting your Research Article entitled 'EPIC: inferring relevant cell types for complex traits by integrating genome-wide association studies and single-cell RNA sequencing' to PLOS Genetics.

The manuscript was fully evaluated at the editorial level and by independent peer reviewers. One reviewer has minor suggestion about the simulation and we would like you to include such simulation results. 

We therefore ask you to modify the manuscript according to the review recommendations. Your revisions should address the specific points made by each reviewer.

[LINK]

Yours sincerely,

Xiaofeng Zhu

Section Editor: Methods

PLOS Genetics

David Balding

Section Editor: Methods

PLOS Genetics

Reviewer's Responses to Questions

**Comments to the Authors:**

Reviewer #1: The authors answered all my concerns and I don't have further questions.

Reviewer #2: My previous comments were all well addressed.

Reviewer #3: The authors have addressed my comments adequately.

One specific comment: while evaluating EPIC's type I error control for its gene-level association testing, the authors chose Bernoulli(0.5) to simulate the case-control phenotype. However, more realistic choices are 0.1, 0.2 in the population. I would suggest expanding the type 1 error rate study for these choices as well.

**Have all data underlying the figures and results presented in the manuscript been provided?**

Reviewer #1: Yes

Reviewer #2: None

Reviewer #3: None

PLOS authors have the option to publish the peer review history of their article (what does this mean?). If published, this will include your full peer review and any attached files.

Reviewer #1: No

Reviewer #2: No

Reviewer #3: No

---

## [Editor Report · Decision Letter 2]

12 May 2022

Dear Dr Jiang,

We are pleased to inform you that your manuscript entitled "EPIC: inferring relevant cell types for complex traits by integrating genome-wide association studies and single-cell RNA sequencing" has been editorially accepted for publication in PLOS Genetics. Congratulations!

Yours sincerely,

Xiaofeng Zhu

Section Editor: Methods

PLOS Genetics

David Balding

Section Editor: Methods

PLOS Genetics

Comments from the reviewers (if applicable):

**Data Deposition**

http://datadryad.org/submit?journalID=pgenetics&manu=PGENETICS-D-21-01527R2

**Press Queries**

---

## [Editor Report · Acceptance letter]

10 Jun 2022

PGENETICS-D-21-01527R2 

EPIC: inferring relevant cell types for complex traits by integrating genome-wide association studies and single-cell RNA sequencing 

Dear Dr Jiang, 

We are pleased to inform you that your manuscript entitled "EPIC: inferring relevant cell types for complex traits by integrating genome-wide association studies and single-cell RNA sequencing" has been formally accepted for publication in PLOS Genetics! Your manuscript is now with our production department and you will be notified of the publication date in due course.

With kind regards,

Zsofia Freund

PLOS Genetics

On behalf of:
